# Potent Therapeutic Strategies for COVID-19 with Single-Domain Antibody Immunoliposomes Neutralizing SARS-CoV-2 and Lip/cGAMP Enhancing Protective Immunity

**DOI:** 10.3390/ijms24044068

**Published:** 2023-02-17

**Authors:** Yajun Zhou, Xing Lu, Xiaoqing Wang, Tianlei Ying, Xiangshi Tan

**Affiliations:** 1Department of Chemistry, Fudan University, 2005 Songhu Road, Shanghai 200438, China; 2Institutes of Biomedical Sciences, School of Basic Medical Sciences, Fudan University, Shanghai 200032, China

**Keywords:** COVID-19, nanobodies, liposomes, cGAMP, immunity adjuvant

## Abstract

The worldwide spread of COVID-19 continues to impact our lives and has led to unprecedented damage to global health and the economy. This highlights the need for an efficient approach to rapidly develop therapeutics and prophylactics against SARS-CoV-2. We modified a single-domain antibody, SARS-CoV-2 VHH, to the surface of the liposomes. These immunoliposomes demonstrated a good neutralizing ability, but could also carry therapeutic compounds. Furthermore, we used the 2019-nCoV RBD-SD1 protein as an antigen with Lip/cGAMP as the adjuvant to immunize mice. Lip/cGAMP enhanced the immunity well. It was demonstrated that the combination of RBD-SD1 and Lip/cGAMP was an effective preventive vaccine. This work presented potent therapeutic anti-SARS-CoV-2 drugs and an effective vaccine to prevent the spread of COVID-19.

## 1. Introduction

The novel coronavirus disease 2019 (COVID-19), caused by severe acute respiratory syndrome coronavirus 2 (SARS-CoV-2), continues to impact our lives and has led to unprecedented damage to global health and the economy [1,2,3]. According to the World Health Organization, as of 28 October 2022, there have been 626,337,158 confirmed cases of COVID-19, including 6,566,610 deaths. By 25 October 2022, a total of 12,830,378,906 vaccine doses had been administered. For preventing the worldwide spread of COVID-19, an efficient approach to rapidly develop therapeutics and prophylactics against SARS-CoV-2 is urgent.

The SARS-CoV-2 spike protein, containing the receptor-binding domain (RBD) and S1 subunit involved in receptor engagement, is a potential therapeutic target. Using the spike (S) glycoprotein on the viral surface, SARS-CoV-2 infection of a human cell is initiated by the recognition of the human membrane protein angiotensin-converting enzyme 2 (ACE2) [4,5]. Based on the first reported genome sequence of 2019-nCoV [1], Wrapp et al. expressed ectodomain residues 1 to 1208 of 2019-nCoV S and obtained the cryo-EM structure of the 2019-nCoV spike in the prefusion conformation [4]. The S protein exists as a homotrimer with each protomer consisting of an N-terminal S1 subunit and a C-terminal S2 subunit, cleaved by host proteases (such as furin) during viral formation [6]. From its metastable prefusion conformation, the S protein undergoes dramatic conformational changes to form a highly stable post-fusion conformation, resulting in shedding of the S1 subunit and virus–host membrane fusion mediated by the S2 subunit [4]. The S1 subunit is composed by an N terminal domain (NTD), a receptor-binding domain (RBD), and two subdomains (SD1 and SD2). Demonstrated by cryo-EM structures, the RBD in the S1 subunit is highly dynamic and adopts “up” (or “open”) and “down” (or “close”) states, where “up” corresponds to receptor-accessible and “down” corresponds to a receptor-inaccessible state [4,5]. The SARS-CoV-2 RBD is less exposed than that of SARS-CoV, which potentially contributes to the immune surveillance evasion and the wide spread of the virus [7]. Researchers can use the SARS-CoV-2 RBD to screen for neutralizing monoclonal antibodies (mAbs) and design vaccines against SARS-CoV-2 [8,9]. Yang et al. have reported that a SARS-CoV-2 RBD-based recombinant COVID-19 vaccine could induce potent neutralizing antibody responses in immunized mice, rabbits, and non-human primates (NHPs), and provided protection in NHPs against SARS-CoV-2 challenges [10]. In this study, we cloned the sequence of a 2019-nCoV RBD-SD1 fragment (S residues 319 to 591) according to the research of Wrapp et al. [4], which bound to ACE2 well with K_D_ = 34.6 nM, and expressed the protein as an antigen for further immune experiments.

A number of neutralizing mAbs were developed and their therapeutic potential was proved in the treatment of coronavirus infections [8,9,11,12,13]. However, their clinical usefulness has been hampered by time-consuming and costly antibody manufacturing processes in eukaryotic systems. Timely manufacturing is difficult in an epidemic setting because the large-scale production of mAbs typically takes at least 3 to 6 months. An attractive alternative for monoclonal antibodies is single-domain antibodies (sdAbs). The sdAbs are the autonomous variable domains of heavy chain-only antibodies and Ig new antigen receptors, which are also known as nanobodies or VHHs if originating from the camelid family (including camels, llamas, and vicugna) [14]. The nanobodies can acquire affinities and specificities for antigens comparable to conventional antibodies, and their small size provides several advantages, including a larger number of accessible epitopes, relatively low production costs, and ease of rapid production at kilogram scale in prokaryotic expression systems [15]. Therefore, sdAbs are becoming a promising alternative to mAbs. Wrapp et al. reported that MERS VHH-55, SARS VHH-72 and VHH-72-Fc may serve both as useful reagents for researchers and as potential therapeutic candidates, because of their potent neutralization capacity for betacoronaviruses [16]. In addition, bivalent SARS VHH-72 neutralized SARS-CoV-2 pseudoviruses [16], while Wu et al. developed a versatile platform for rapid isolation of fully human single-domain antibodies and for screening of antibodies against SARS-CoV-2 [17]. n3088, n3130 and n3113 were identified that could neutralize SARS-CoV-2 pseudoviruses [17]. These antibodies could represent promising candidates for prophylaxis and the therapy of COVID-19. In this study, we modified the SARS VHH-72 sequence [16] and expressed the protein in *E. coli*. For the soluble overexpression, the small ubiquitin-like modifier tag 3 (Smt3) tag was added into the sequence. The overexpression protein with the tag was named Smt3-SARSVHH, and after small ubiquitin-like modifier (SUMO) protease cleavage, the purified protein was named SARSVHH. Both of the proteins were linked to the liposomes, respectively, TVHH-Lip and VHH-Lip. We hoped that the liposomes linked by nanobodies could neutralize viruses and also carry antivirals, loaded in the liposomes, to the area of infection. The immunoliposomes would be a potent therapy of COVID-19.

The endoplasmic reticulum–resident adaptor protein stimulator of interferon (IFN) genes (STING) is a key signaling molecule that is activated after cytosolic DNA detection. 2′,3′-Cyclic guanosine monophosphate–adenosine monophosphate (cGAMP), a natural agonist of STING, is a secondary messenger generated in response to DNA viral infections or tissue damage [18,19]. Upon DNA binding in the cytosol, cyclic GMP-AMP synthase (cGAS) is activated and produces the cGAMP [20]. As a second messenger, cGAMP combines with the STING of endoplasmic reticulum protein, then STING recruits and activates TBK1 (TANK in combination with kinase 1), which phosphorylates IRF3 (interferon regulatory factor 3), inducing IFN-I (type I interferon) production and a series of inflammatory immune responses [21]. IFN-Is initiate immediate a cell intrinsic antiviral response to contain the pathogen, as well as an adaptive immune response to achieve long-term protection [22]. Li et al. performed high-throughput screening and identified endogenous STING agonists, e.g., cGAMP, as antivirals against SARS-CoV-2 [23]. Moreover, STING agonist may serve as a novel therapeutic strategy to combat COVID-19, treating infection therapeutically against diverse strains of SARS-CoV-2 [23]. While cGAMP has low potency and serves as a poor drug, potent small-molecule STING agonists have been developed [24]. However, cGAMP still has unique properties of being a small molecule (dinucleotide) and a potent activator of STING signaling, and it has been demonstrated as a potent adjuvant to elicit Ab and T cell responses in mice [25]. The challenge is the cytosolic delivery of naked cGAMP, because of the inherent dual negative charges of cGAMP and the presence of an extracellular enzyme that cleaves cGAMP [26]. At present, many nanomedicines related to the STING pathway have been produced and have realized effective cytoplasmic delivery of the antigen presenting cells (APCs) [27,28]. To overcome the delivery limitations, we have developed a liposome strategy that encapsulates cGAMP, with modifications to reduce systemic distribution, improve the recognition ability of target cells, and increase the intracellular drug dose. In our previous study, the liposome strategy enhanced the STING agonist activity and improved the efficiency of tumor therapy via the cGAMP-STING-IRF3 pathway [29]. 

In this research, we designed and prepared cGAMP-loaded liposomes. The surface of liposomes was modified with Smt3-SARSVHH nanobody (TVHH-Lip/cGAMP) or SARSVHH nanobody (VHH-Lip/cGAMP). We hoped that the immunoliposomes could neutralize viruses and also carry the antiviral, cGAMP, to the area of infection. Moreover, we prepared cGAMP-loaded liposome, Lip/cGAMP. STING agonists are potent adjuvants capable of eliciting robust antitumor immunity after intratumoral administration and augmenting intradermal influenza vaccines [30,31]. Lip/cGAMP was used as a potent adjuvant, which was combined with the antigen RBD-SD1 as a recombinant vaccine. Our aim was that the recombinant vaccine would be more effective and timely responded by the immune system, with the adjuvant Lip/cGAMP enhancing immunity.

## 2. Results and Discussion

### 2.1. Preparation and Characterization of Nanobodies and Immunoliposomes

The nanobody Smt3-SARSVHH was successfully expressed and purified. After SUMO protease cleavage of Smt3-SARSVHH, the purified protein SARSVHH was obtained. Smt3-SARSVHH and SARSVHH nanobodies were assessed by 15% SDS-PAGE (Figure 1A). The molecular weights of the nanobodies were about 26 kD for Smt3-SARSVHH, and 14 kD for SARSVHH. The proteins were found to be at least 95% pure based on the denaturing polyacrylamide gels stained with Coomassie Brilliant Blue (Figure 1A).

The structure of TVHH-Lip/cGAMP and VHH-Lip/cGAMP is shown in Figure 1B. The drug encapsulation efficiency (EE%), particle size, zeta potential, polydispersity index (PDI) and drug loading efficiency (LE%) are listed in Table 1.

cGAMP was hydro soluble, and the ammonium sulfate gradient method was selected to prepare liposomes. The hydrogenated soybean lecithin was selected to improve the stability of liposomes at high temperature, and the final drug encapsulation efficiency was above 80%. The particle size of the non-targeted liposomes was about 180 nm, and the particle size of the targeted liposomes was slightly increased due to the connected proteins on the external surface. The mean diameter was generally accepted within an optimal size range for a liposome drug delivery system. Anionic surface charge was also considered preferable for intratumoral distribution of liposome, while minimizing non-specific uptake by cell types other than APCs [32,33]. The polydispersity index (PDI) of liposome formulations was less than 0.2, indicating that the liposomes were homogeneous and could be used for drug administration. The zeta potential of liposomes was negative due to the lecithin and proteins. It is worth noting that the weak negative charge was favorable for a long circulation to enhance the accumulation of nanodrugs [34]. The morphology of cGAMP-loaded liposomes was studied by transmission electron microscopy (TEM). The cGAMP-loaded liposomes were uniformly spherical, with a diameter of 170~210 nm (Figure 1C). Moreover, no significant difference was observed in the EE% and particle size of liposomes stored at 4 °C after 3 weeks, confirming that the prepared liposomes had good physical stability.

The concentrations of phospholipids and nanobodies in the liposome sample were calculated and are shown in Table 2.

### 2.2. Neutralization of Nanobodies and Immunoliposomes

We first measured the neutralization activities of the nanobodies Smt3-SARSVHH and SARSVHH with the pseudovirus neutralizing assay (Figure 2B). TVHH and VHH are short for Smt3-SARSVHH and SARSVHH, respectively. The VHH sequence was modified from SARS VHH-72 [16]. Bivalent SARS VHH-72 neutralized SARS-CoV-2 pseudoviruses [16]. The modified Smt3-SARSVHH and SARSVHH demonstrated the neutralization activities with the pseudovirus neutralizing assay. Smt3-SARSVHH and SARSVHH had IC50 values of 9.75 and 1.916 μg/mL (0.375 and 0.136 μM), respectively (Figure 2C). The sequences of the nanobodies are list in Appendix A.

TVHH-Lip/cGAMP and VHH-Lip/cGAMP showed moderate neutralization activities, inhibiting SARS-CoV-2 pseudovirus infection in a dose-dependent manner with half-maximal inhibitory concentration (IC50) values of 20.6 and 10.59 μg/mL (0.792 and 0.756 μM), respectively (Figure 2A,C). Obviously, Lip/cGAMP was not linked with the nanobody, and did not have the ability to neutralize SARS-CoV-2 pseudoviruses (Figure 2A). We developed two neutralizing SARS-CoV-2 pseudoviruses nanobodies. Moreover, the immunoliposomes TVHH-Lip/cGAMP and VHH-Lip/cGAMP showed moderate neutralization activities, and cGAMP was an antiviral which could be delivered to the infection focus. Both immunoliposomes would be potent therapies against SARS-CoV-2.

### 2.3. Preparation and Characterization of Antigen RBD-SD1 and Lip-cGAMP

We cloned the sequence of a 2019-nCoV RBD-SD1 fragment (S residues 319 to 591) (Figure 3A) according to the research of Wrapp et al. [4]. For the soluble overexpression, the Smt3 tag was added into the sequence. The overexpression protein with the tag was named Smt3-RBD-SD1, and after SUMO protease cleavage, the purified protein was named RBD-SD1. The protein RBD-SD1 was successfully expressed and purified. The purified protein RBD-SD1 was assessed by 15% SDS-PAGE (Figure 3B). The molecular weight of RBD-SD1 was about 30 kD and the purity was more than 95% based on the denaturing polyacrylamide gels stained with Coomassie Brilliant Blue (Figure 3B).

cGAMP was synthesized by the cGAS enzymatic synthesis method [35] with the existence of ATP, GTP, and some metal salt in specified conditions. Lyophilized endotoxin-free cGAMP was stored at −20°C for subsequent use. The mass spectrum data regarding successful cGAMP synthesis were inferred in Figure 4A. cGAMP was hydrosoluble, and the ammonium sulfate gradient method was selected to prepare liposomes. The final drug encapsulation efficiency was above 80% (Table 1). The particle size of the Lip/cGAMP was about 180 nm (Figure 4C). The prepared cGAMP-loaded liposome was broken by solvent (methanol: isopropanol: ddH_2_O = 14:6:5, *v*/*v*/*v*), and the cGAMP concentration was determined with a standard curve by HPLC (Figure 4B). The morphology of Lip/cGAMP was also studied by transmission electron microscopy (TEM). Lip/cGAMP was uniformly spherical, with a diameter of 170~200 nm (Figure 4D). For Lip/cGAMP safety, we examined the blood routine of the normal mice and Lip/cGAMP addressed mice (20 mg/kg). The results of normal mice and Lip/cGAMP addressed mice showed no significant difference (Appendix A).

### 2.4. Imunization of Mice with the Recombinant Vaccine RBD-SD1 and Lip-cGAMP

Mice were immunized with a standard dose of 30 μg RBD-SD1 as the antigen and different doses of Lip/cGAMP (1 eq cGAMP, 2 eq cGAMP, 4 eq cGAMP vs. the dose of antigen). RBD-SD1 with empty liposomes was used as a control group, and the dose of empty liposomes was the same as 2eq cGAMP. RBD-SD1 with the unpacked cGAMP (4eq vs. the dose of antigen) was used as another control group. cGAMP can boost the immune response without the liposomes, but the unpacked cGAMP without liposomes has poor bioavailability, as cGAMP could be hydrolyzed in the cytosolic delivery. It was demonstrated that a high dose of cGAMP was required to achieve the obvious enhancement of immune response in the early pre-experiments. Mice were sensitized on multiple spots with antigen in complete Freund adjuvant on day 0 and sera were collected on day 14. The challenge immunization was the same to immunize on multiple spots with antigen in incomplete Freund adjuvant, and sera were collected on day 21. The sera collected on day 14 (Prime) or 21 (Boost) were measured for Ag-specific IgG (Figure 5A). Then, the sera collected on day 21 were tested for the neutralizing antibody levels with the 2019-nCoV Surrogate Virus Neutralization kit (ATaGenix, Wuhan, China). RBD protein was pre-coated on the enzyme label plate. The neutralizing antibody of the RBD in the serum samples would block the interaction between the RBD and the ACE2. The unbonded reagent was washed, and the chromogenic substrate solution was added to the well for the color development. The color intensity was inversely proportional to the RBD neutralizing antibody concentration in the serum samples. It was demonstrated that all groups of the RBD-SD1 immunization produced the neutralizing antibody (Figure 5B).

At day 14, Lip/cGAMP groups already had the immune response (Figure 5C). At day 21, the serum IgG titers were measured (Figure 5D). The group with a high dose of Lip/cGAMP demonstrated an enhanced immune response.

Lip/cGAMP showed the ability of enhancing protective immunity. Groups with a high dose of Lip/cGAMP obtained a higher neutralizing antibody level. With the adjuvant Lip/cGAMP, the recombinant vaccine was induced timely, and the immune response was more effective.

## 3. Materials and Methods

### 3.1. Prepration of Nanobodies and RBD-SD1

The sequences of proteins were designed according to the literature [4,16,36,37]. Coding sequences were subcloned into the *E. coli* periplasmic expression vector pET-22b(+) with 6 His-tag in the C-terminal. Rosetta *E. coli* containing the plasmid were grown at 37 °C in LB plus ampicillin. After inducing with 1 mM IPTG at 30 °C for 16 h, the cells were harvested by centrifugation, resuspended in buffer (50 mM Tris, 300 mM NaCl, pH = 8.0), and lysed by ultrasonic. The periplasmic fraction was isolated by centrifugation at 10,000 g for 20 min at 4 °C and then loaded onto Ni-NTA in buffer (50 mM Tris, 300 mM NaCl, 10 mM imidazole, pH = 8.0). Proteins were eluted by a gradient to 50 mM Tris, 300 mM NaCl, 500 mM imidazole, 10% glycerin, pH = 8.0. The eluted proteins were loaded onto a Superdex S-200 column to increase purity. Recombinant nanobodies were assessed by 15% SDS-PAGE. The purified proteins were stored at −80 °C.

To remove bacterial endotoxin, proteins were immobilized on HisTrap HP 5 mL column (GE Healthcare) in PBS. The proteins were washed with PBS containing 0.1% Triton X-114, and eluted with LPS-free PBS with 500 mM imidazole. Imidazole was removed by dialysis. LPS concentration was measured using an LAL Chromogenic Endotoxin Quantitation Kit (Thermo Fisher Scientific, Waltham, MA, USA), and all proteins were LPS purified (<2 IU/mg).

### 3.2. Synthesis of Nanobody-Modified Liposomes

To prepare nanobody-targeted liposomes (TVHH-Lip/cGAMP or VHH-Lip/cGAMP), the nanobody was modified with the sulfhydryl group according to the previous method [38]. Briefly, Traut’s reagent (2-iminothiolane) was added to the prepared nanobodies solution drop-by-drop (protein: 2-iminothiolane = 1: 20, molar ratio), and the proteins were incubated for 1 h at 37 °C. The free thiolation agent was removed by dialysis.

Liposomes were prepared by the ammonium sulfate gradient method. TVHH-Lip/cGAMP or VHH-Lip/cGAMP was composed of HSPC, cholesterol, mPEG_2000_-DSPE and DSPE-PEG_2000_-Mal (9:3:2.4:0.6, *w*/*w*). Briefly, lipids above were dissolved in chloroform to form a thin lipid film by rotary evaporation at 40 °C. A 250 mM ammonium sulfate solution was added to the phospholipid membrane at 65 °C to form multilayer liposomes. The liposomes were extruded through 200 nm pore size polycarbonate filter (Avastin) to obtain the monolayer empty liposomes with uniform particle size, which were dialyzed to remove ammonium sulfate solution in 5% glucose at 4 °C.

The cGAMP solution was added to empty liposomes, and incubated at 60 °C for 1 h. Unencapsulated drugs were removed by dialysis in 5% glucose. Modified nanobodies were added to drug-carrying liposomes (lipids: protein = 50:1, mass ratio) and incubated overnight at RT in the dark. Free nanobodies were removed by dialysis in 5% glucose. The unmodified control liposomes were prepared with the same protocol as described earlier, except for the addition of DSPE-PEG_2000_-Mal and nanobodies. For long-term storage, liposomes could be freeze-dried with 10% trehalose and stored at −20 °C.

### 3.3. Characterization of cGAMP-loaded Liposomes

The prepared cGAMP-loaded liposome was broken by solvent (methanol: isopropanol: ddH_2_O = 14:6:5, *v*/*v*/*v*), and the cGAMP concentration was determined with a standard curve by HPLC. After vigorous vortex mixing and centrifugation, the water phase was injected into an isocratic 20:80 (*v*/*v*) water/methanol mobile phase, operating at 1.0 mL/min, 31 °C column temperature, 260 nm detector setting. After passing through a C18 column (Agilent), the absorbance of the eluent was measured at 260 nm. A sample resulting chromatogram is attached in Appendix A. Drug encapsulation efficiency (EE%) was calculated by the following equation: $$\mathrm{EE}\%=\frac{cGAMP_{en}}{cGAMP_{tol}}\times 100\%$$
where cGAMP_en_ is the mass of encapsulated cGAMP, and cGAMP_tol_ is the total mass of cGAMP.

Drug loading efficiency (LE%) was calculated by the following equation:$$\mathrm{LE}\%=\frac{cGAMP_{mass}}{liposome_{mass}}\times 100\%$$
where cGAMP_mass_ is the mass of encapsulated cGAMP, and liposomes_mass_ is the total mass of loaded liposomes.

The average particle size distribution, polydispersity index (PDI), and zeta potential of nanobody-targeted liposomes were measured by dynamic light scattering (Zetasizer Nano-ZS90, Malvern Panalytical Ltd., Malvern, UK). The morphologies of liposomes were examined by a transmission electron microscope (Tecnai G2 F20S-Twin, FEI Ltd., Portland, OR, USA).

To evaluate the loading stability, we stored the liposomes at 4 °C for a period of time. The particle size and cGAMP encapsulation efficiency were measured by DLS and ultrafiltration methods, respectively.

A 25 μL liposome sample was taken and 4 mL chloroform was added, followed by 4 mL of reagent (1.3515 g ferric chloride, 1.52 g ammonium thiocyanate, 50 mL ddH_2_O). The mixture was centrifugated at 4000 rpm for 10 min. The upper layer was removed, and the absorption of the lower layer was measured by an ultraviolet spectrophotometer at 485 nm. According to the standard curve, the phospholipid concentration of liposome samples was calculated.

Nanobodies connected on the surface of liposomes were extracted by the methanol–chloroform oscillation extraction method. The concentration of extracted proteins was determined by a BCA kit, and 15% SDS-PAGE gel electrophoresis was used to confirm the band location and purity of the proteins.

### 3.4. Pseudotyped Virus Neutralization

To determine the neutralization activity of single-domain antibodies, a pseudotyped virus neutralization assay was performed. Briefly, 293 T cells were co-transfected with expression vectors of pcDNA3.1-SARS-CoV-2-S (encoding SARS-CoV-2 S protein) and pNL4-3.luc.RE bearing the luciferase reporter-expressing HIV-1 backbone, as previously described [39]. The supernatants containing SARS-CoV-2 pseudotyped virus were harvested. Serial dilutions of single-domain antibodies in DMEM supplemented with 10% fetal calf serum were incubated with pseudoviruses at 37 °C for 1 h and then the mixtures were added to monolayer Huh-7 cells (10^4^ per well in 96-well plates). Twelve hours after infection, the culture medium was refreshed and then incubated for 48 h. The luciferase activity was calculated for the detection of relative light units using the Bright-Glo Luciferase Assay System (Promega). A nonlinear regression analysis was performed on the resulting curves using Prism (GraphPad) to calculate half maximal inhibitory concentration (IC50) values.

### 3.5. Enzyme-Linked Immunosorbent Assay (ELISA)

Costar half-area high binding assay plates (Corning #3690) were coated with purified protein at 100 ng/well in PBS overnight at 4 °C, and blocked with PBS buffer containing 3% milk powder (*w*/*v*) at 37 °C. For polyclonal phage ELISA, phages from each round of panning were incubated with immobilized antigen, and bound phages were detected with anti-M13-horseradish peroxidase (HRP) polyclonal antibody (Pharmacia). For the purified antibody binding assay, serially diluted antibody solutions were added and incubated for 1.5 h at 37 °C, and bound antibodies were detected with monoclonal anti-Flag-HRP antibody (Sigma-Aldrich). The enzyme activity was measured with the subsequent addition of substrate ABTS (Invitrogen) and signal reading was carried out at 405 nm using a Microplate Spectrophotometer (Biotek).

### 3.6. Immunize mice and Antibody Measurement

Mice (BALB/c, male, 6–8 weeks of age) were immunized with a standard dose of 30 μg RBD-SD1 as the antigen and together with different doses of Lip/cGAMP (1 eq cGAMP, 2 eq cGAMP, 4 eq cGAMP vs. the dose of antigen). Empty liposomes served as a control group, and the dose of empty liposomes was the same as 2 eq cGAMP. Mice were sensitized on multiple spots with antigen in complete Freund adjuvant on day 0 and sera were collected on day 14. The challenge immunization was the same to immunize on multiple spots with antigen in incomplete Freund adjuvant. Sera were collected on day 21. To assess the humoral immune responses induced by the recombinant RBD-SD1, we used an enzyme-linked immunosorbent assay (ELISA) for RBD-specific antibodies. The upper serum layer was collected and stored at −20 °C. RBD-SD1 (or S2 protein as a control) was used to coat flat-bottom 96-well plates (Corning #3690) at a final concentration of 1 μg/mL in 50 mM carbonate coating buffer (pH 9.6) at 4 °C overnight. The following day, plates were washed three times with PBS containing 0.1% Tween-20 (PBST), and blocking solution containing 1% BSA in PBST was added, followed by 1 h incubation at room temperature. Serially diluted mouse sera were added and incubated at 37 °C for 1 h, and then the plates were washed three times with PBST. Antibodies, including goat anti-mouse IgG horseradish peroxidase (HRP)-conjugated antibody and anti-mouse IgG1/IgM HRP-conjugated antibody, were diluted 1:5000 in blocking solution and added to the wells (100 μL/well). After incubation for 1 h at room temperature, the plates were washed five times with PBST and developed with 3,3′,5,5′-tetramethylbiphenyldiamine (TMB) for 10 min. The reactions were stopped with 50 μL/well of 1.0 M H_2_SO_4_ stop solution. The absorbance was measured on a microplate reader at 450 nm. To measure the titer of RBD-specific antibodies induced by recombinant proteins, serum samples were serially diluted and measured by titration.

The neutralizing antibody levels were tested with the 2019-nCoV Surrogate Virus Neutralization kit (ATaGenix, Wuhan, China). This kit was based on the neutralizing ELISA analysis technology. RBD protein was pre-coated on the enzyme label plate. The positive controls, negative controls, and analysis samples were added to the wells for reaction, and then the HRP-labeled ACE2 protein was added. The RBD neutralizing antibody in the samples would block the interaction between the RBD and the ACE2. The unbonded reagent was washed, and the chromogenic substrate solution was added to the well for the color development. The color intensity was inversely proportional to the RBD neutralizing antibody concentration in the sample. RBD neutralizing antibodies in serum and plasma samples were determined.

## 4. Conclusions

In this study, we developed two neutralizing SARS-CoV-2 pseudoviruses nanobodies, Smt3-SARSVHH and SARSVHH, with IC50 values of 9.75 and 1.916 μg/mL (0.375 and 0.136 μM), respectively. Moreover, the nanobodies were modified to the Lip/cGAMP. Two immunoliposomes, TVHH-Lip/cGAMP and VHH-Lip/cGAMP, were synthesized. Both immunoliposomes could neutralize the SARS-CoV-2 pseudoviruses, the IC50 values were 20.6 and 10.59 μg/mL (0.792 and 0.756 μM), respectively. Smt3-SARSVHH and SARSVHH would be potent therapeutic drugs against SARS-CoV-2 because of their neutralization activities. The difference between Smt3-SARSVHH and SARSVHH was the Smt3 tag. SARSVHH had a lower IC50 value than Smt3-SARSVHH. It might be that the Smt3 tag prevented the effective part binding to the S protein of the virus. The Smt3 tag is 11 kd, and the SARSVHH is 14 kd. The molecules’ sizes are similar between the tag and the SARSVHH. This could be the reason for the lower binding affinity of Smt3-SARSVHH, such that the effective part of Smt3-SARSVHH would contact the S protein less. There was a risk that the nanobodies’ epitopes would be blocked when the nanobodies were linked to the liposomes. In addition, the flexibility of nanobodies would be influenced by the liposomes. However, the moderate neutralization activities of TVHH-Lip/cGAMP and VHH-Lip/cGAMP demonstrated that the linked nanobodies were still effective. The immunoliposomes could also be therapeutic drugs against SARS-CoV-2, because of their neutralizing ability. Although the neutralization activities of the immunoliposomes were lower than the nanobodies’, they were targeted liposomes loaded with antivirals. The antiviral, cGAMP, would be mostly delivered to the infected area. With the targeted liposomes, the hydrolysis of antivirals in the cytosolic delivery and the dispersion of drug distribution were avoided. 

The nanobodies and immunoliposomes could be a potent therapy against SARS-CoV-2. We hoped that the nanobodies and immunoliposomes would be effective toward other different COVID-19 strains. Furthermore, novel nanobodies and other antivirals could employ the strategy of the immunoliposome. 

In the prophylactic strategy for preventing the spread of COVID-19, we developed an effective and timely vaccine. We used the 2019-nCoV RBD-SD1 protein as an antigen with Lip/cGAMP as the adjuvant to immunize mice. STING agonists were potent adjuvants capable of eliciting robust antitumor immunity. It was demonstrated that a much higher dose of cGAMP was required to achieve the obvious enhancement of immune response than liposome-packed cGAMP. Lip/cGAMP was an ideal potent adjuvant which could enhance the protective immunity. RBD-SD1 as the antigen was used to determine serum IgG titers by ELISA. The produced antibodies after the RBD-SD1 immunizations were tested with the 2019-nCoV Surrogate Virus Neutralization kit. RBD-SD1 as the candidate vaccine antigen could maintain most neutralization sensitive epitopes [4]. The antibodies after the RBD-SD1 immunizations showed the neutralizing ability to SARS-CoV-2. In the Prime phase, the groups of RBD-SD1 with Lip/cGAMP demonstrated the immune response. The RBD-SD1 group and the RBD-SD1 with Lip group had the same level of the immune response. It was demonstrated that Lip/cGAMP would induce the immune system timely. In the Boost phase, the high dose groups of RBD-SD1 with Lip/cGAMP demonstrated a more powerful immune response and a higher antibody level. It was demonstrated that Lip/cGAMP would enhance the immune system significantly.

The combination of RBD-SD1 and Lip/cGAMP was an effective preventive vaccine. Coronavirus could variate and the S protein would have various variants. The potent adjuvant Lip/cGAMP employed in vaccines would prevent the spread of the epidemic. 

In summary, two strategies against COVID-19 were developed in this study: one being a potential therapeutic strategy of anti-SARS-CoV-2 immunoliposomes to cure the disease and the other being an effective vaccination method with Lip/cGAMP as the adjuvant to prevent the spread of COVID-19.

## Figures and Tables

**Figure 1 ijms-24-04068-f001:**
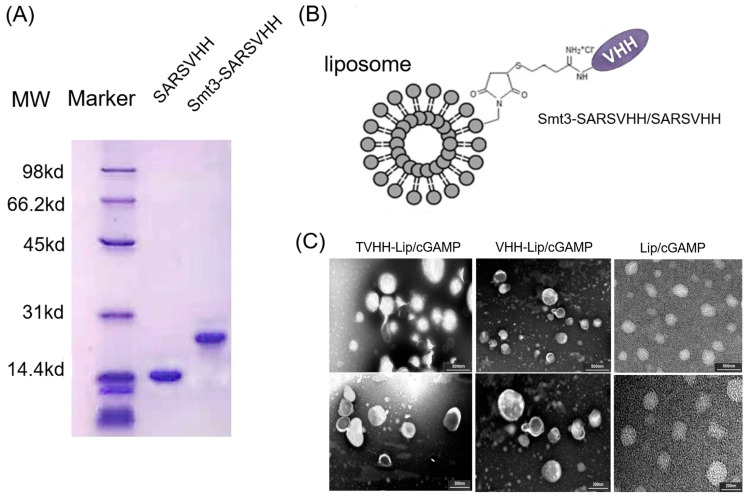
(**A**)The purified nanobodies SARSVHH and Smt3-SARSVHH. (**B**) The structure of TVHH-Lip/cGAMP and VHH-Lip/cGAMP. (**C**) The TEM revealed cGAMP-loaded liposomes with a uniform spherical shape (upper lane scale bars: 500 nm, lower lane scale bars: 200 nm).

**Figure 2 ijms-24-04068-f002:**
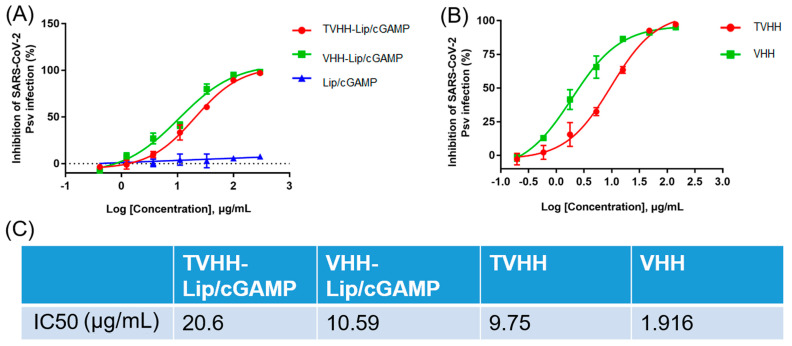
Neutralization activities of immunoliposomes (**A**) and nanobodies (**B**). The IC50 values (**C**) of the neutralization.

**Figure 3 ijms-24-04068-f003:**
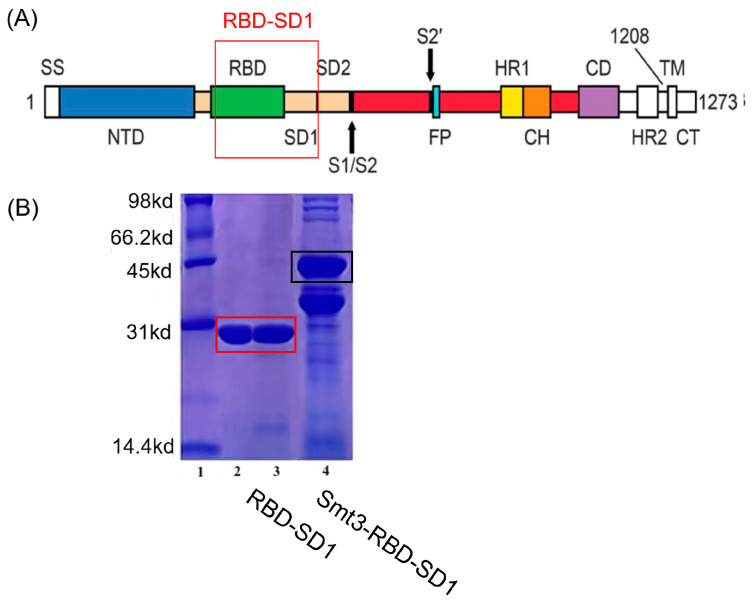
(**A**) A schematic view of the SARS-CoV-2 spike protein. (**B**) The purified protein RBD-SD1 in lane 2 and 3.

**Figure 4 ijms-24-04068-f004:**
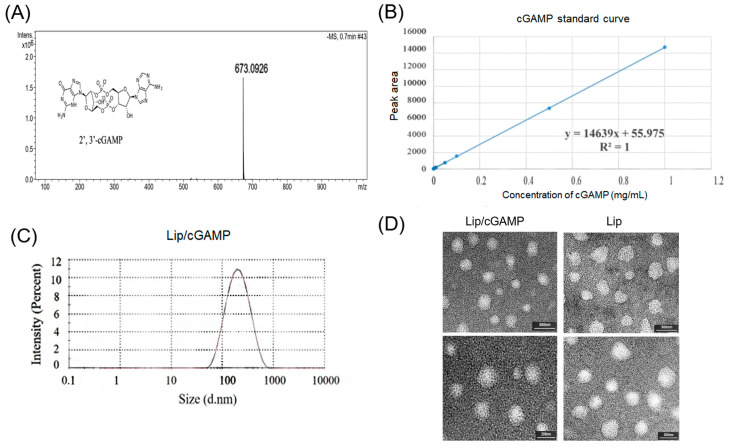
(**A**) The chemical structure and mass spectrum of the 2′,3′-cGAMP molecule (**B**) The standard curve of cGAMP. (**C**) The particle size distribution of Lip/cGAMP. (**D**) The TEM revealing Lip/cGAMP and empty liposomes (Lip) with a uniform spherical shape (upper lane scale bars:500 nm, lower lane scale bars: 200 nm).

**Figure 5 ijms-24-04068-f005:**
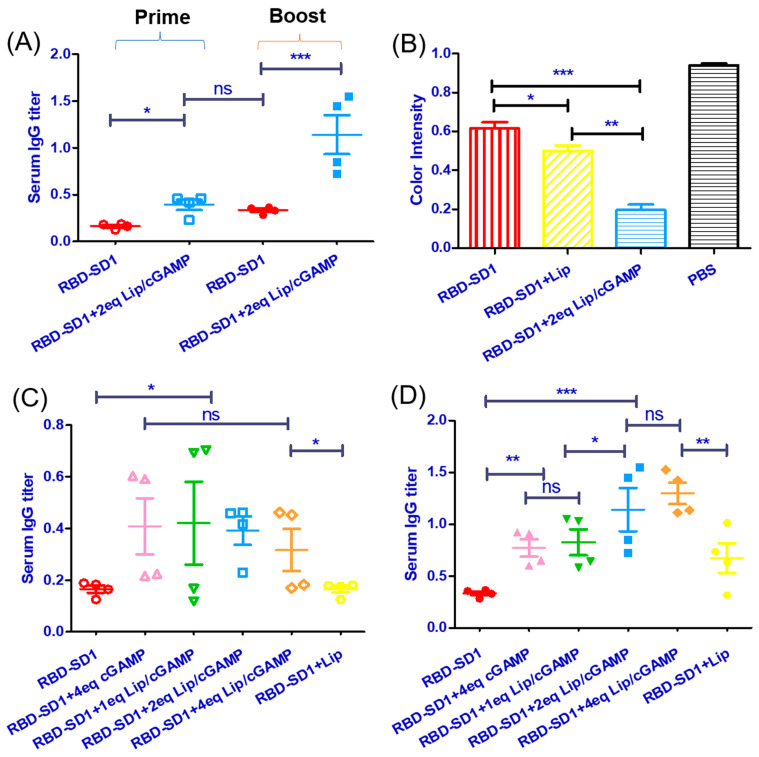
(**A**) Sera were collected on day 14 (Prime) or 21 (Boost) and measured for Ag-specific IgG. (**B**) Neutralizing antibody levels were determined with the 2019-nCoV Surrogate Virus Neutralization kit. Color intensity was inversely proportional to the RBD neutralizing antibody levels. (**C**) At day 14, the serum IgG titers were measured. (**D**) At day 21, the serum IgG titers were measured. *n* = 4 mice. Statistical analysis, one-way ANOVA * *p* < 0.05, ** *p* < 0.01, and *** *p* < 0.001, ns: no significance. All experiments were repeated twice with similar results. Each symbol represents individual mice. Red ○/●: group RBD-SD1 in Prime/Boost phase; pink △/▲: group RBD-SD1+ 4eq cGAMP in Prime/Boost phase; green ▽/▼: group RBD-SD1+ 1eq Lip/cGAMP in Prime/Boost phase; blue □/■: group RBD-SD1+ 2eq Lip/cGAMP in Prime/Boost phase; orange ◇/◆: group RBD-SD1+ 4eq Lip/cGAMP in Prime/Boost phase; yellow ⎔/⬣: group RBD-SD1+ Lip in Prime/Boost phase.

**Table 1 ijms-24-04068-t001:** The characterization of cGAMP-loaded liposomes.

Formulation	Particle Size (nm)	Zeta Potential (mV)	PDI	EE (%)	LE (%)
Lip/cGAMP	175.63 ± 0.73	−0.42 ± 1.17	0.185	83.42 ± 1.73	8.33 ± 0.16
TVHH-Lip/cGAMP	204.73 ± 4.21	−13.30 ± 1.20	0.182	83.13 ± 2.58	8.35 ± 0.21
VHH-Lip/cGAMP	204.53 ± 3.81	−12.80 ± 2.50	0.189	83.47 ± 6.79	8.34 ± 0.19

**Table 2 ijms-24-04068-t002:** The composition of nanobody targeted liposomes.

Formulation	Phospholipid Concentration (mg/mL)	Protein Concentration (μg/mL)	Protein Density (μg Protein/mg Phospholipid)
TVHH-Lip/cGAMP	11.87 ± 0.38	263.70 ± 32.52	20.20 μg/mg
VHH-Lip/cGAMP	12.71 ± 0.82	278.72 ± 22.42	20.53 μg/mg

## Data Availability

The data presented in this study are available upon request from the corresponding authors.

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
