# Peer review of "Potent Therapeutic Strategies for COVID-19 with Single-Domain Antibody Immunoliposomes Neutralizing SARS-CoV-2 and Lip/cGAMP Enhancing Protective Immunity"

_ijms, 2023, doi:10.3390/ijms24044068_

Round 1
Reviewer 1 Report
This article introduced a method to modify the surface of liposomes with nanobodies as a way to neutralize the SARS-CoV-2 viruses. The paper is well written. I only have some minor suggestions:
1. Figure 2 (A) (B), Figure 4 (A), (B), (C), and Figure 5 (A) (B) (C). Their resolution is too low. Please increase the figure quality, e.g. higher resolution and larger font when needed.
2. The citation sometimes are included as part of the sentence (see line 141) but sometimes are excluded (see line 25). The authors should keep using the same style for the article.
3. The IC50 are normally reported in the unit of micro-molar or nano-molar. Using micro-gram/ml (as shown in Fig 2c) makes it difficult to compare the effect among different immunoliposomes and nanobodies, as the mass is unclear here. Please change the unit.
4. The SI, author contribution, institutional review board statement, etc. should be completed.
5. Based on the current presentation, one gets an impression that TVHH and VHH (nanobody) has a lower IC50 than the immunoliposomes. Readers may wonder why the nanobodies cannot be applied directly to neutralize the viruses. What's the advantage to put them on the surface of liposomes. This part should be clarified.
Reviewer 2 Report
In this paper, the author developed a vaccination method using 2019-nCoV RBD-SD1 protein as specific antigen and Lip/cGAMP as adjuvant, aiming at developing therapeutic and preventive drugs against SARS-CoV-2, which is theoretically feasible. However, the data in this paper lack some logic and consistency. Many important experiments have not been well proved clearly, and the writing of the article also needs to be carefully revised. I will elaborate from the following aspects:
1、In Fig5, RDB-SD should be RDB-SD1, and there is no coordinate unit in FIG5. The level of fig5c Lip group and RDB-SD1 group is basically the same, and it is doubtful whether specific antibodies can be produced.
2、The text should test for neutralizing antibody levels, not just IGg levels.
3、If the authors want to use Lip/cGAMP as a carrier and adjuvant, I think the immunogenicity, safety and RBD bearing capacity of Lip/cGAMP should also be tested.
4、Does the reason for co-expressing the SD1 domain increase the antibody level after immunization?
5、Please directly show the information of the strain from spike and the strain information based on which the fake virus is constructed. Is VHH effective against different COVID-19 strains?
6、How to explain the higher IC50 of VHH after connecting cCAMP?
7、The results of different parts of the literature were poorly correlated. The authors demonstrated the antiviral effect of VHH. Can it be used to prepare chimeric vaccine of RBD?
8、It is suggested to increase the discussion and expound the experimental results, research prospect and value.
Reviewer 3 Report
Zhou et al. contributed a research article entitled “Potent therapeutic strategies to COVID-19 with Single-Domain 2 antibody immunoliposomes neutralizing SARS-CoV-2 and 3 Lip/cGAMP enhancing protective immunity” was very well written and clearly structured. However, after reviewing the manuscript, I do see some gaps, that author should fulfil before publication.
My comments are as below
1. Please elaborate on morphology difference between empty liposome and cGAMP loaded liposome by TEM.
2. Explain on size difference of empty liposome, non-targeted and targeted liposome by TEM.
3. Line 168 and Lines 171-173, How did you calculate IC50 values? Can you explain in detail the results you got for Figure 2? I see in methods you have wrote 3 line (325-327) that mentions how you got the results, but I think you may need to give further explanations how you got IC50 values (May be calculations).
4. Line 283. Author has briefly explained about how concentration of cGAMP liposomes quantified by HPLC. I think author must need to mention clear method conditions for HPLC experiment (Column temperature, detector setting, and resulting chromatogram should attach in supplementary) So it will be easier for anyone to read and refer the paper who specifically looking for quantification cGAMP liposomes by HPLC.
Reviewer 4 Report
In this paper, the authors proposed 2 novel liposome/cGAMP-based strategies as countermeasures against SARS-CoV-2. The ideas are innovative, but the results were poorly presented with lots of grammar errors. I believe with more time, these results will turn out to be 2 nice papers. However, both of the stories are incomplete and the manuscript is not acceptable in its present form. Here are some general suggestions to hopefully improve this paper. For the first part (VHH-Lip/cGAMP), more evidence is needed to verify the liposomes can be directed to virus-infected cells through VHH and that the cargo in the liposome can be delivered into the target cell. Also, it would be nice to include some data from in vivo studies to further evaluate the efficacy of the immuno-liposomes. For the second part (Lip/cGAMP as adjuvant), IgG titer is not convincing by itself, and more experiments, like serum neutralization assay/cytokine level analysis/virus challenge assays, should be performed to extensively evaluate how Lip/cGAMP promotes the immune response in animal models.
Major comments:
1. There are lots of grammar errors in the paper. I would suggest having the manuscript carefully proofread by a native English speaker.
2. Figure 1B: Please replace the IgG with a VHH molecule in the diagram.
3. Figure 1C: the TEM images seem to be distorted and I don’t think 3-4 liposomes are representative. Could you please provide the original TEM images, preferably in different magnifications? It would be nice to add control images of liposomes without VHH attaching, so we can see if VHH changes the shape/diameter of the liposomes.
4. I assume the Spike protein from the original Wuhan strain was used in the neutralization assay. What are the neutralizing potencies of VHH-72 against other variant strains?
5. Figure4: please improve the resolutions of the figures in A, B, and C.
6. Figure5: please add titles for Y axes and swap the order of B and C. Statistical analysis is lacking in these figures. Which antigen was used to determine serum IgG titers by ELISA?
Minor comments:
1. Line 80: Please specify what “SUMO” stands for.
2. Line 81: “Both of two proteins” should be “Both of the proteins”.
3. Line 90: Please specify what “cGAS” stands for.
4. Line 104: “The challenging is” should be “The challenge is”.
5. Line 109: “with modification” should be “with modifications”
6. Line 121: change “more quickly” to “timely”.
7. Lines 129&190: please specify how did you quantify the protein purity here?
8. Line 147-148: the particle diameters were determined by dynamic light scattering. But in this sentence, it looks like they were determined by TEM images. Please rephrase this sentence to avoid confusion.
9. Line 167: remove “However,”.
10. Line 174: “linked” should be “linked with”
11. Line 246: change “broken” to “lysed”.
Round 2
Reviewer 2 Report
Thank you very much for your reply to all my questions.
Author Response
Thanks a lot for your kind suggestions and nice comments.
Reviewer 4 Report
1. The figures for Lip/cGAMP in 1C and 4D are the same. You should either use 2 different sets of images in each figure or keep them in only one figure.
2. The particles of VHH-Lip/cGAMP look quite different from the Lip/cGMP. Was that caused by different staining conditions?
3. I don't think there is much improvement in the grammar errors. Especially in the discussion part, some sentences don't make sense at all.
4. Lack of information (strain/gender/age) for the mice study.
5. I wonder if cGAMP can boost the immune response without the liposome, as cGAMP is a good adjuvant by itself. I would recommend including another control that mice injected with RBD+unpacked cGAMP.
6. My biggest concern was not addressed in the revision. The manuscript still looks like sewing 2 unfinished studies together instead of one complete story.
Round 3
Reviewer 4 Report
I'm ok with the manuscript, but please have a native English speaker proofread the paper again.
Author Response
The manuscript was checked and the English writing was improved in the revised version. Thanks a lot for your kind suggestions and nice comments.